# Prognostic and Predictive Molecular Markers in Cholangiocarcinoma

**DOI:** 10.3390/cancers14041026

**Published:** 2022-02-17

**Authors:** Sandra Pavicevic, Sophie Reichelt, Deniz Uluk, Isabella Lurje, Cornelius Engelmann, Dominik P. Modest, Uwe Pelzer, Felix Krenzien, Nathanael Raschzok, Christian Benzing, Igor M. Sauer, Sebastian Stintzing, Frank Tacke, Wenzel Schöning, Moritz Schmelzle, Johann Pratschke, Georg Lurje

**Affiliations:** 1Department of Surgery, Campus Charité Mitte, Campus Virchow Klinikum, Charité—Universitätsmedizin Berlin, 13353 Berlin, Germany; sandra.pavicevic@charite.de (S.P.); sophie.reichelt1@charite.de (S.R.); deniz.uluk@charite.de (D.U.); felix.krenzien@charite.de (F.K.); nathanael.raschzok@charite.de (N.R.); christian.benzing@charite.de (C.B.); igor.sauer@charite.de (I.M.S.); wenzel.schoening@charite.de (W.S.); moritz.schmelzle@charite.de (M.S.); johann.pratschke@charite.de (J.P.); 2Department of Gastroenterology and Hepatology, Campus Charité Mitte, Campus Virchow Klinikum, Charité—Universitätsmedizin Berlin, 13353 Berlin, Germany; isabella.lurje@charite.de (I.L.); cornelius.engelmann@charite.de (C.E.); frank.tacke@charite.de (F.T.); 3Department of Hematology, Oncology and Cancer Immunology, Campus Charité Mitte, Campus Virchow Klinikum, Charité—Universitätsmedizin Berlin, 13353 Berlin, Germany; dominik.modest@charite.de (D.P.M.); uwe.pelzer@charite.de (U.P.); sebastian.stinzing@charite.de (S.S.)

**Keywords:** cholangiocarcinoma, biliary tract cancer, biomarker, prognosis, predictive, targeted therapy

## Abstract

**Simple Summary:**

Cholangiocarcinoma (CCA) is a heterogenous and aggressive malignancy of the intra- and extrahepatic biliary tract, marked by a steeply rising incidence on a global scale. While surgery remains the only curative treatment option, most patients present with advanced or unresectable disease, and are, therefore, treated with systemic therapy, albeit with limited benefit. Biomarkers obtained from either the patients’ serum or tumor tissue might facilitate therapy guidance by selecting patients who would benefit the most from surgical and adjuvant treatment strategies, as well as by identifying those with higher risk of disease recurrence. Furthermore, several genetic aberrations in CCA have been linked with improved response upon targeted therapies, thus highlighting their role as predictive biomarkers. In this review we provide an overview of currently known prognostic and predictive biomarkers and discuss their role in CCA.

**Abstract:**

Cholangiocarcinoma (CCA) is the second most common primary liver cancer and subsumes a heterogeneous group of malignant tumors arising from the intra- or extrahepatic biliary tract epithelium. A rising mortality from CCA has been reported worldwide during the last decade, despite significant improvement of surgical and palliative treatment. Over 50% of CCAs originate from proximal extrahepatic bile ducts and constitute the most common CCA entity in the Western world. Clinicopathological characteristics such as lymph node status and poor differentiation remain the best-studied, but imperfect prognostic factors. The identification of prognostic molecular markers as an adjunct to traditional staging systems may not only facilitate the selection of patients who would benefit the most from surgical, adjuvant or palliative treatment strategies, but may also be helpful in defining the aggressiveness of the disease and identifying patients at high-risk for tumor recurrence. The purpose of this review is to provide an overview of currently known molecular prognostic and predictive markers and their role in CCA.

## 1. Introduction

Cholangiocarcinoma (CCA) is a highly aggressive malignancy and the second most common primary liver tumor, accounting for three percent of all gastrointestinal malignancies [1]. The classification is based upon the anatomical site of origin, thus differentiating between intrahepatic (iCCA), perihilar (pCCA) (also called Klatskin tumor), and distal (dCCA) cholangiocarcinoma, with dCCA and pCCA being summarized as extrahepatic cholangiocarcinomas (eCCA). The most common entities are dCCA and pCCA, accounting for about 30% and 50% of all CCAs, respectively. In contrast, iCC represents only about 10% of all CCAs [2]. The anatomical boundary defining eCCA is the cystic duct, with dCCA arising distal and pCCA proximal of the junction of the cystic duct, comprising the right and left hepatic ducts up to the second order biliary branches. Tumors located above the second order bile ducts are termed iCCA [1,3,4]. Different types of CCA do not only differ in their etiology, pathophysiology, and treatment, but also possess unique biological and pathological features, providing the opportunity for individualized prognosis determination and targeted therapies.

Many CCA cases are sporadic, arising in the absence of known risk factors such as chronic biliary inflammation (e.g., in primary sclerosing cholangitis—PSC), cholestasis, hepatobiliary parasitic infections (liver flukes like *Clonorchis sinensis* and *Opisthorchis viverrini*), and liver cirrhosis (Figure 1) [1]. Furthermore, emerging evidence over the past years has suggested an important role of the gut microbiome on development of CCA, especially in patients with inflammatory bowel disease and PSC [5]. Epigenetic and environmental factors seem to have a significant impact on the development the disease, since there is a great difference in prevalence of CCA between Southeast Asia (e.g., 85 per 100,000 in Northeast Thailand) and the Western hemisphere (<6 per 100,000 population). Symptoms of CCA are usually vague and arise late in already-advanced disease, resulting in median survival of less than two years from the timepoint of diagnosis. So far, surgery remains the only curative treatment option, with a 5-year survival rate ranging between 25 and 50% after surgical resection [6,7,8,9,10,11]. Due to high risk of recurrence, especially in patients with lymph node metastasis, tumor-positive resection borders, and low-grade CCA, adjuvant chemotherapy is recommended by current guidelines [12,13,14]. Patients with metastatic or inoperable disease are treated with systemic chemotherapy, albeit with dismal benefit [7,15,16,17]. Advancements in organ preservation and poor survival outcomes following surgical resection have promoted consideration of liver transplantation (LT) as a curative approach for patients suffering from pCCA and iCCA [18,19]. In fact, LT in patients adhering to the so-called Mayo clinic protocol, compromising neoadjuvant chemoradiation in patients with unresectable pCCA, size <3 cm and without extrahepatic or lymph node metastasis, offered promising results [20]. A French multicenter randomized, intent-to-treat study (NCT02232932) comparing the 5-year survival following capecitabine-based chemoradiotherapy with subsequent LT to standard liver resection in patients suffering pCCA is ongoing [21]. In contrast to pCCA, iCCA is a contraindication in most centers worldwide due to historically poor outcome. However, recent data suggest LT as an effective treatment in highly selected patients with localized and early iCCA or patients with disease stability after neoadjuvant chemotherapy [11]. Results from a multicenter, single-arm, prospective study (NCT02878473) evaluating the 5-year survival in patients with single iCC ≤ 2 cm in size, liver cirrhosis, and CA 19-9 ≤ 100 ng/mL, undergoing LT are eagerly awaited. While the number of patients suffering from CCA continues to rise, the need for novel diagnostic strategies and therapeutic options has become of utmost clinical importance. As such, the identification of preoperatively available molecular markers of prognosis as an adjunct to traditional staging systems has emerged as a promising strategy to select patients who may benefit the most from surgical and adjuvant therapy. Furthermore, the identification of predictive markers might result in precise and effective systemic treatment.

## 2. Biomarkers in CCA

Biomarkers can have either prognostic or predictive value, while certain biomarkers are known to possess both. Prognostic biomarkers inform about the likelihood of certain cancer-associated events, e.g., disease-recurrence or progression of disease, and overall survival. Predictive biomarkers provide information about treatment benefit and are used to identify individuals who are more likely to have a favorable or unfavorable effect from a particular therapy compared to individuals without the biomarker [22,23,24]. The utility of molecular markers for clinical practice has been assessed by defining their levels of evidence, ranging from level I as the highest evidence level, obtained from high-powered, prospective, randomized controlled trials (RCT), to level V, which possess the weakest evidence and are derived from single clinical cases. Importantly, in order for a novel biomarker to be implemented into the routine clinical practice, at least level II evidence is required [25].

Different types of CCA do not only show distinct anatomical and histological features, but also possess individual molecular profiles and genetic aberrations. In this context, Nakamura et al. identified a total of 32 significantly altered genes in about 40% of cases by molecularly characterizing biliary tract cancers from 260 patients [26]. Another research group analyzed 410 cancer-associated genes in tumor samples of 195 patients and discovered genetic alterations with potential therapeutic implications in 47% these of patients [27]. The spectrum of actionable genomic targets in CCA embraces various kinases (FGFR1, FGFR2, FGFR3, PIK3CA, ALK, EGFR, ERBB2, BRAF and AKT3), oncogenes (IDH1, IDH2, CCND1, CCND3 and MDM2) and tumor-suppressor genes (BRCA1 and BRCA2) [26]. The most common genetic alterations shared between iCCA and eCCA are KRAS mutations (15%–20%), TP53 mutations (15 %–25%) and AT-rich interactive domain-containing protein 1A (ARID1A) mutations (approx. 20%) [28]. Certain alterations, such as FGFR1/2 fusions or IDH1 mutations, are unique to iCCA and rarely occur in eCCA. Mutations in BRCA1 associated protein 1 (BAP1) are also enriched in iCCA and present in less than 1% of eCCA. In contrast, mutations in ERBB2 seem to be exclusive to eCCA [26,29]. An overview of common signaling pathways involved in the development and progression of CCA is displayed in Figure 2.

In the following paragraphs, we will provide an overview of the most relevant biomarkers used for prognosis prediction in patients with CCA. Additionally, currently known molecular aberrations, which were identified as predictive biomarkers of treatment response, will be discussed.

## 3. Prognostic Serum Biomarkers

Prognostic markers in patients with CCA are currently based on clinical factors, such as tumor extent, existence of metastasis, surgical resection margin or histological tumor differentiation. Over the last years, various molecular markers of prognosis, obtained mainly from peripheral blood and tumor tissue, have been proposed for the detection and prediction of prognosis in CCA (Table 1).

### 3.1. Serum Proteins

So far, carbohydrate antigen (CA19-9) and carcinoembryonic antigen (CEA) are the most widely used biomarkers for diagnosis and surveillance of CCA. CA 19-9, a glycoprotein mainly produced by biliary and pancreatic duct cells, has been associated with poor prognosis, while preoperatively elevated levels of CA 19-9 proved to be a negative independent prognostic factor in CCA [31,32]. Furthermore, CA 19-9 decline ≥ 50% under chemotherapy with gemcitabine correlated with improved therapy response and increased survival in patients with advanced CCA [33]. However, the sensitivity and specificity of this biomarker is 72% and 84%, respectively, thus limiting its diagnostic and prognostic value [34]. CEA is a commonly used tumor marker in colorectal cancer but has evolved as a relevant biomarker in CCA as well [35]. In fact, studies have reported high variations in terms of sensitivity and specificity of CEA in patients with CCA, ranging from approximately 40%–80% and 50%–90%, respectively [33,36,37,38]. CEA has been identified as an independent prognostic marker in CCA, especially in combination with CA 19-9 [36,38,39,40,41,42]. Nonetheless, the partially low sensitivity and specificity reported should be kept in mind when using this biomarker for diagnosis and surveillance of CCA. Other promising serum diagnostic and prognostic biomarkers include cytokeratin-19 fragment (CYFRA 21-1), matrix metalloproteinase-7 (MMP-7) and osteopontin. CYFRA 21-1 showed a negative correlation with one-year outcome in patients with iCCA and gallbladder carcinoma, but no association with eCCA was found [43]. Interestingly, CYFRA 21-1 and MMP-7 levels were both elevated in patients with CCA compared to those with benign biliary disease, emphasizing their putative diagnostic value [44]. The role of osteopontin, a secreted extracellular matrix glycophosphoprotein, remains inconclusive. While Loosen et al. showed that serum osteopontin levels are increased in patients with CCA and associated with impaired survival, another study demonstrated the opposite [45,46]. Although data on these biomarkers seem promising, further studies are required to validate current findings.

### 3.2. Inflammatory Biomarkers

Circulating cytokines have been associated with disease progression, tumor stage or treatment response in numerous malignancies. In the context of CCA, increased levels of interleukin-6 (IL-6) were found in patients with CCA, compared to healthy individuals, showing a rather moderate sensitivity (73%) and specificity of 92%. Furthermore, high expression of IL-6 in serum and tumor tissue, as well as marked expression of interleukin-17 (IL-17) in peritumoral cells negatively correlated with overall survival (OS) and disease-free survival (DFS) in patients who underwent surgical resection of iCCA [47].

The urokinase plasminogen activator receptor (suPAR) is an inflammatory mediator and the soluble form of the cell surface receptor uPAR (CD87). suPAR was demonstrated to be an independent prognostic biomarker in various cancer types. In terms of CCA, increased uPAR expression in tumor tissue and suPAR expression in patients’ serum has been associated with impaired survival, whereas another study showed a positive correlation between uPAR expression and lymphatic invasion and metastasis, respectively [48,49]. Furthermore, elevated baseline levels of uPAR were predictive of poor survival in patients treated with palliative chemotherapy due to inoperable CCA [50].

Since a combination of two prognostic markers may improve the power of prognosis, the ratio of certain serum proteins with high diagnostic and predictive power was applied in several cancer types. As such, various ratios of myeloid and lymphatic cells have been proposed as prognostic markers in CCA. Indeed, the neutrophil-to-lymphocyte ratio, a well know prognostic factor in gastric and lung cancer, was associated with impaired survival in patients with resected CCA, as well as in those under chemotherapy due to advanced disease. Moreover, C-reactive-protein to albumin ratio negatively correlated with OS and DFS in CCA [51]. The albumin to gamma-glutamyltransferase ratio (AGR) proved to be an independent prognostic indicator for iCCA following curative resection, demonstrating improved predictive accuracy compared with the TNM staging alone [52]. Other inflammatory biomarker ratios calculated from peripheral blood measurements are still under investigation. While relatively simple to determine, these putative outcome-predicting ratios have not become part of the clinical routine yet.

### 3.3. Circulating Nucleic Acids

In recent years, circulating nucleic acids, such as cell-free DNA (cfDNA) or RNA, mostly microRNA (miRNA), have emerged as promising biomarkers for detection and prognosis prediction in multiple cancer types due to their abundance and stability in biofluids. In the case of CCA, the sensitivity and specificity of pooled miRNAs were calculated at up to 80% and 90%, respectively, in several metanalyses [53,54,55,56]. Notably, miRNA measured in bile samples showed the highest diagnostic efficiency. Increased serum and plasma levels of miR-21 positively correlated with the TNM stage and poor survival and decreased after surgical resection of the tumor. Notably, estimation of miR-21 enabled differentiation between patients with CCA and healthy controls, however miR-21 was increased in other malignancies as well, thus limiting its specificity [57]. Further studies investigating other miRNAs are controversial. While reduced levels of miR-150 were observed in individuals with CCA, another study reported an upregulation of miR-150 in patients suffering iCCA. Remarkably, the combination of reduced miR-150 and increased CA19-9 seemed to improve the accuracy of CCA diagnosis [58]. Expression of miRNAs was further investigated in bile samples and correlated with CCA occurrence; however, their prognostic value remains inconclusive [57].

cfDNA represents a fragment of DNA which is released upon cell apoptosis or necrosis, a common process in tumorigenesis. The most appealing fact about cfDNA determination is the ability to screen for overall mutation patterns of the respective malignancy without the need for obtaining primary tumor tissue. To support these findings, plasma samples from 31 patients with CCA were screened for oncogenic mutations. The results showed that the same mutation patterns could be observed in the tumor itself [59]. Hence, cfDNA might facilitate the detection of specific genomic alterations and subsequently the establishment of effective, mutation-based therapies.

### 3.4. Single-Nucleotide Polymorphisms

Single-nucleotide polymorphisms (SNPs) are genetic modifications defined by a substitution of a single nucleotide at a specific position in the genome. Depending on the function of the affected genetic region, SNPs have been associated with clinical outcomes and cancer susceptibility in a large variety of malignancies, making them potential prognostic or therapeutic targets [60,61,62,63,64]. Most recently, a single-center analysis of multiple genes involved in tumor inflammation and angiogenesis revealed CXCR1 (interleukine-8-receptor alpha—IL-8RA) +860 C>G heterozygous polymorphism to be an independent prognostic factor for DFS, cancer-specific survival and OS in patients with pCCA [65]. Further studies have linked SNPs to CCA. The G protein subunit-β 3 (GNB3) 825 C>T polymorphism was associated with longer OS in patients with eCCA [66]. Other variants, such as the enhancer of zeste homolog 2 (EZH2), nuclear factor (erythroid-derived 2)-like 2 (NRF2), x-ray repair cross-complementing group (XRCC1), ATP binding cassette subfamily C member 2 (ABCB2), ATPase Phospholipid Transporting 8B1 (ATP8B1), natural killer cell receptor G2D (NKG2D), and alpha1-antitrypsin (α1AT) deficiency Z heterozygosity, have been linked to increased risk of CCA or associated with the outcome of patients with bile duct tumors [64,67,68]. Nevertheless, biomarker-embedded clinical trials and validation in independent cohorts of patients are required to confirm these preliminary findings.

### 3.5. Other Biomarkers with Potentially Prognostic Value

Circulating tumor cells (CTC), various metabolites found in bile, blood, or urine, as well as extracellular vesicles represent some of other biomarkers currently evaluated for diagnosis and outcome prediction in patients with CCA [69]. While metabolites and extracellular vesicles have been investigated mostly for diagnostic purpose and differentiation between CCA and other hepatic malignancies or non-malignant biliary diseases, CTC have shown an association with DFS and OS in numerous malignancies. However, the number of studies suggesting that CTCs may have an impact on clinical outcome in CCA are scarce [57].

Taken together, numerous molecular markers of prognosis in CCA with promising results from clinical studies have been identified over recent years. Before implementation into clinical routine, larger patient cohorts and biomarker-embedded clinical trials are needed. Furthermore, novel technologies enabling the measurement of cf-DNA or miRNAs, will facilitate the detection of novel biomarkers in the future. Even though determination of CA19-9 and CEA has certain limitations, they remain the most frequently used and best evaluated serum biomarkers in CCA.

## 4. Prognostic Tumor Tissue Biomarkers

Biomarkers from tumor tissue present a valuable source of potential factors for outcome prediction in terms of both survival and treatment response, or individualized therapies in patients suffering CCA (Table 2). In fact, CCA was described as a highly genomic heterogeneous malignancy, with most of the genetic alterations being related to DNA repair mechanism, chromatin remodeling or cancer cell proliferation and growth [57]. While some mutations are specific to either iCCA or eCCA, others are found in tumor tissue irrespective of the anatomical localization. The latter include mutations in the KRAS proto-oncogene (15–22%), TP53 tumor suppressor gene (25–40%), ARID1A chromatin remodeling complex (12–18%), and BRCA1/2 (3–5%). Mutations in KRAS and TP53 have been associated with impaired outcome and tumor recurrence following surgical resection of CCA, thus highlighting their prognostic value [70]. The role of ARID1A as a prognostic marker remains inconclusive since it has been described as both an oncogene and tumor-suppressor gene. Inactivation of ARID1A correlated with tumor metastasis and was common in liver fluke-associated CCA [71]. Another study confirmed the prognostic role of ARID1A by showing a correlation between low ARID1A expression and impaired outcome in patients with iCCA [72]. In contrast, increased expression of ARID1A was associated with a higher risk of mortality and disease recurrence in patients with iCCA [73]. In a large metanalysis, including 4,126 patients from 73 studies, that analyzed 77 known biomarkers, EGFR, MUC1, MUC4, fascin, and p27 showed an association with OS of patients suffering from CCA [74]. Another metanalysis assessed prognostic biomarkers associated with OS in patients with eCCA in a univariate analysis. While six markers (VEGF, COX-2, GLUT-1, cyclin D1, fascin, and Ki-67) correlated with impaired survival, p16, p27, and E-Cadherin had positive prognostic effects [75]. Moreover, the analysis of 53 patients with surgical tumor resection due to CCA revealed 39 transcriptomic prognostic biomarkers. Interestingly, all of them showed a relation with T-cell activation and immune response. For instance, the expression levels of cytotoxic T-lymphocyte antigen 4 (CTL4) and forkhead box P3 (FOXP3) correlated with recurrence-free survival [76].

### 4.1. Cell Surface Molecules

Several cell surface molecules are known to have significant impact on cancer progression by regulating cell motility and transcellular signaling. As such, expression of CD155, an immunoglobulin-like transmembrane glycoprotein, was associated with shorter DFS and OS in patients suffering CCA. Upregulation of CD155 correlated with histological grading, lymph node metastasis, expression of vascular endothelial growth factor (VEGF), and microvascular density, and was suggested as an independent prognostic marker for CCA [77]. High expression of CD44 correlated with significantly shorter OS compared to low intratumoral expression of CD44 in patients with liver fluke-associated CCA [78]. Furthermore, CD55 and CD97 showed an association with poor histological grading, lymph node metastasis, venous invasion, and shorter OS, while CD98 was proposed as an independent prognostic factor in CCA [79].

### 4.2. Signaling Molecules

Diverse signaling molecules, mainly cytokines and intracellular signaling molecules, are directly involved in carcinogenesis and have been associated with patients’ outcome in multiple cancer types. As such, increased levels of IL-6 in tumor tissue and IL-17 in peritumoral cells correlated with impaired OS and DFS in patients with iCCA. Furthermore, multivariate analysis revealed that IL6 and peritumoral IL17 are independent prognostic factors for DFS, while preoperatively increased levels of IL-6 in serum were associated with significantly reduced DFS [47]. Suppressor of cytokine signaling 3 (SOCS3) is an antagonist of the JAK/STAT pathway, thus playing an integral role in shaping the inflammatory environment and tumorigenesis in CCA. The expression of SOCS3 was significantly downregulated in tumor tissue of patients with CCA, while the upstream regulator tumor necrosis factor α-induced protein 3 (TNFAIP3 or A20) was increased. Notably, patients with low intratumoral expression of SOCS3 and high expression of A20 showed a dramatically reduced OS rate. Moreover, both proteins were associated with lymph node metastasis and postoperative disease recurrence, thereby suggesting their role as prognostic markers in CCA [80].

Ring finger protein 43 (RNF43), which has been described as both an oncogene and tumorsuppressor gene, was downregulated in tumor tissue of patients with iCCA and correlated with poor prognosis. Furthermore, RNF43 was shown to be an independent prognostic factor in uni- and multivariate analysis [81].

LIM and SH3 protein 1 (LASP-1) is a focal adhesion protein, known to play a key role in cell migration, invasion, and proliferation in a wide variety of tumors. Analysis of human CCA tissue samples revealed that LASP-1 was markedly overexpressed in tumor compared to healthy tissue. Moreover, expression of LASP-1 positively correlated with tumor size, poor histological differentiation, lymph node metastasis, advanced TNM stage, and poor prognosis in CCA patients. In contrast, downregulation of LASP-1 resulted in cancer cell apoptosis and suppressed cell migration, invasion, and proliferation in vitro [82].

Similarly, the expression of B7-H4, which is a member of the B7 superfamily of ligands and regulator of T cell-mediated antitumor immune response, was upregulated in cancer tissue in approximately 50% of cases in a cohort of 137 patients suffering from CCA and was associated with poor histological differentiation, lymph node metastasis, staging, reduced OS, and early recurrence of tumor. Additionally, B7-H4 suppressed the peritumoral infiltration of CD8+ cytotoxic T lymphocytes [83]. Hepatoma-derived growth factor (HDGF) is another biomarker that has been associated with poor outcome and tumor progression in patients suffering CCA [84]. High expression of Ki-67, a well-known proliferation marker, and p73 was associated with shorter OS of pCCA patients, while Ki-67 correlated with tumor stage [85].

Induction of epithelial mesenchymal transition (EMT) results in increased cancer cell proliferation and metastasis. Indeed, SOX4 transcription factor, a member of the SOX (SRY-related HMG-box) family, has been shown to promote EMT in vitro, while SOX4 overexpression in tumor tissue indicated poor prognosis in patients with iCCA [86]. Moreover, elevated serum level of nardilysin (N-arginine dibasic convertase, NRDC), a soluble cytosolic protein, correlated with increased NRDC mRNA expression in tumor tissue and EMT-inducing transcription factors, and was associated with shorter OS and DFS in patients with iCCA [87].

### 4.3. Mucins

Mucins (MUC) are heavily O-glycosylated proteins, mainly expressed by ductal and glandular epithelial tissues. Vast production of mucus is frequently found in various carcinomas and has been described in CCA multiple times. In this context, MUC5AC was aberrantly expressed in tumor tissue and associated with larger tumor size and advanced stage in liver fluke-associated CCA [88]. Another study showed an association between MUC5AC expression in iCCA tissue and lymph node metastasis in patients who underwent curative-intent hepatectomy. Additionally, increased MUC5AC expression was identified as an independent prognostic marker of poor survival in patients with iCCA [89].

### 4.4. Tumor Stroma and Microenvironment

The tumor microenvironment plays a crucial role in shaping the growth, proliferation, and metastasis of malignant cells. Thus, certain molecules localized in tumoral stroma might be used as prognostic markers. Indeed, overexpression of epithelial cell adhesion molecule (EpCAM) in the stroma of ICC proved to be an independent risk factor for OS and DFS. Besides expression in iCCA tissue, overexpression of EpCAM in nontumor fibrous liver tissue correlated with reduced DFS as well [90]. Furthermore, high expression of Lysyl oxidase-like 2 (LOXL2), a matrix-remodeling enzyme that has already been associated with metastasis in hepatocellular carcinoma (HCC), in peritumoral stroma proved to be an independent prognostic factor of worse OS and DFS in patients with iCCA [91]. The metalloproteinases (MMP) are enzymes that degrade extracellular matrices, thus facilitating tumor cell progression and metastasis. MMP-9 and MMP-11 were markedly expressed in CCA tissue, and their expression correlated with decreased OS [92,93]. Tissue expression of MMP-9 was associated with IL-8 tissue expression, while the latter proved to be an independent prognostic factor of OS in patients with CCA [94].

### 4.5. Non-Coding RNA

The emerging role of serum-derived non-coding RNAs as prognostic biomarkers has been described in the previous section. Similarly, non-coding RNA, mostly microRNA (miRNA), small interfering RNA (siRNA), and long non-coding RNA (lncRNA), can be measured in tumor tissue. In this context, different studies reported an overexpression of lncRNA H19 and lncRNA-PANDRA in CCA tissue, as well as a significant correlation with tumor progression, TNM staging and OS of patients. Accordingly, in vitro analysis demonstrated the involvement of both RNAs in cell growth and proliferation, EMT and anti-apoptosis [95,96]. Hence, lncRNA H19 and PANDRA may serve as poor prognostic markers for CCA.

Using a custom microarray, the expression levels of three miRNAs, miR-675-5p, miR-652-3p and miR-338-3p, were identified in CCA tissue and strongly correlated with the prognosis of patients with iCCA. Risk scores, defined by using regression coefficients risk, revealed significantly shorter OS and DFS medians in patients with high-risk scores compared to those with low-risk scores. This three-miRNA signature was marked as an independent prognostic predictor for iCCA [97]. Upregulation of tissue miR-29a was suggested as a beneficial prognostic marker for CCA [98].

In conclusion, tumor tissue biomarkers may be of particular value for resected CCA, as they can have both prognostic (e.g., KRAS, TP53), as well as predictive (e.g., IDH1/2, FGFR2) implications by predicting the individual patient’s prognosis and response to targeted therapies. The major drawback remains the necessity of an invasive procedure for tissue sample collection, especially in patients with primary unresectable disease. Alternatively, a preliminary study by Ikeno et al. suggested non-invasive assessment of tumor metabolic activity by ^18^F-fluorodeoxyglucose positron emission tomography (^18^F-FDG-PET) to be associated with impaired survival in patients with iCCA and KRAS mutations [99]. In comparison, serum biomarkers are less invasive to obtain, thus being more suitable for diagnosis and monitoring of disease recurrence. The ideal combination of both biomarker types, as well as sampling timepoints are yet to defined. The detection of genetic aberrations in CCA tissue renders the development of custom and more effective therapies possible and will be discussed in the following paragraphs.

## 5. Predictive Biomarkers of Treatment Response and Novel Therapeutic Strategies

While patients with metastatic or locally advanced non-resectable disease are treated to with palliative chemotherapy, surgical resection with adjuvant chemotherapy represents the gold standard in patients with resectable CCA. According to current guidelines, first-line adjuvant chemotherapy should be conducted with capecitabin, whereas gemcitabin/cisplatin is recommended for patients with advanced disease [100,101,102]. However, the survival rates in both patient groups remain poor, thus prompting the development of targeted and more effective therapies (Supplementary Appendix A). CCA is marked by a high rate of genomic alterations. Even though the most common ones, such as KRAS, TP53 and ARID1A are not easily druggable, >40% of CCA still bear a targetable genetic alteration [28].

### 5.1. Fibroblast Growth Factor Receptor

Fibroblast growth factor receptor 2 (FGFR2) fusions are unique to iCCA and represent one of the main druggable targets. Fibroblast growth factor receptor tyrosine kinases (FGFR) play a key role in activating important signaling pathways, such as RAS-RAF-MEK-ERK and PI3K-AKT-mTOR cascade, thus regulating cellular growth, survival, and differentiation [103,104]. Although the prognostic value of FGFR2 remains controversial, clinical studies evaluating FGFR inhibitors as targeted therapy for patients with advanced CCA show promising results [103,105,106,107,108,109,110,111]. Mazzaferro et al. published in 2019 a multicenter, phase I/II, open-label study exploring safety and effectivity of derazantinib, a pan-FGFR inhibitor, in patients with FGFR2 gen fusion-positive iCCA [112]. With an overall response rate of 20.7% and a disease control rate of 82.8%, derazantinib offered a promising anti-tumor activity, resulting in initiation of a larger study for this patient population (NCT03230318) [112]. Based on a promising phase II trial reporting a 14.8% response rate with a 75.4% disease control rate (NCT02150967; [113,114,115]), the selective FGFR1-3 inhibitor infigratinib is currently evaluated in comparison to gemcitabine and cisplatin in patients with advanced or metastatic CCA and FGFR2 gene fusion in a running phase III RCT (NCT03773302) [116]. Futibatinib and pemigatinib are further selective FGFR kinase inhibitors under current investigation in phase III RCT (NCT04093362 [117]; NCT03656536 [118]).

### 5.2. C-met-Encoded Receptor for Hepatocyte Growth Factor

Another signaling pathway, impacting cell proliferation, motility, and sensitivity to apoptotic cell death, that may be impaired in CCA, is the c-met-encoded receptor (MET) for hepatocyte growth factor signaling pathway [105]. MET is a heterodimeric tyrosine kinase transmembrane protein and receptor of the hepatocyte growth factor [119]. Overexpression of MET occurs in up to 60% of iCCA, whereas mutations in MET axis can be detected in 7% of iCCA [105]. In 2013 Pant et al. examined the combination of tivantinib, an inhibitor of the c-MET tyrosine kinase, and gemcitabine in patients with solid tumors, among them CCA, showing promising results in terms of antitumor activity [120]. In 2017 the research group Goyal et al. demonstrated in a phase II study the limited activity and toxicity of cabozantinib, another multikinase inhibitor targeting MET and vascular endothelial growth factor receptor 2 (VEGFR2) [121]. Further studies are still ongoing (NCT02496208, NCT02711553).

### 5.3. Tyrosine Kinases

Tyrosine kinases are enzymes responsible for the activation of signal transduction cascades through phosphorylation of participative proteins, a step that is inhibited by tyrosine kinase inhibitors. Representative members of the tyrosine kinase family are HER2/neu, VEGFR, platelet-derived growth factor receptor (PDGFR), and FGFR2, all playing an important role in tumorigenesis, cancer progression, and survival. In this context, recent studies have highlighted their use as targeted therapies in treatment of CCA [122]. EGFR is a transmembrane protein of the ErbB tyrosine kinase receptor family, compromising four distinct membrane receptors: EGFR (ERBB1), HER2/neu (ERBB2), ERBB3, and ERBB4 [123]. Overexpression of EGFR and HER2/neu has been detected in up to 30% of iCCA, resulting in promoted tumor cell proliferation, migration, and angiogenesis [123]. In fact, multivariate analysis showed that EGFR expression was a significant prognostic factor and risk factor for tumor recurrence in CCA [124]. Consequently, one could imagine that inhibition of EGFR tyrosine kinase activity might results in decreased progression of CCA, thus making it a promising therapeutical approach. Drugs targeting EGFR can be divided in tyrosine kinase inhibitors, like gefitinib and erlotinib, and monoclonal antibodies, such as cetuximab and panitumumab [125]. Various HER2/neu and EGFR inhibitors are currently being studied, either as single-agents, in addition to established chemotherapy or in combination with other targeted agents, such as the VEGF inhibitor bevacizumab [126,127,128]. Even though initial trials investigating the use of anti-EGFR and anti-HER2/neu inhibitors in patients with advanced CCA were promising [128,129], data from subsequent phase II RCTs were disappointing. In fact, neither the application of lapatinib (EGFR and HER2/neu double inhibitor), panitumumab (anti-EGFR antibody), nor cetuximab (anti-EGFR inhibitor), as monotherapy or in addition to traditional chemotherapy (gemcitabine and oxaliplatin), significantly increased the OS or progression free survival (PFS) [130,131,132,133,134]. Interestingly, the addition of erlotinib, a drug targeting EGFR by tyrosinase kinase inhibition, to chemotherapy with gemcitabine and oxaliplatin significantly prolonged PFS in a subgroup of patients suffering from CCA [135]. Furthermore, the combination of erlotinib with the VEGF antibody bevacizumab showed promising results for treatment of advanced biliary cancer in a multicenter phase II clinical trial [127]. HER2/neu overexpression was found to be higher in eCCA than in iCC [136]. However, studies investigating the use of HER2/neu antibodies, such as pertuzumab and trastuzumab, did not provide convincing results in terms of oncological outcome [137,138]. Additional studies investigating targeted therapies directed at HER2/neu or the use of multikinase inhibitors are ongoing.

### 5.4. Angiogenesis

VEGF and its receptors play a pivotal role in tumor angiogenesis. In fact, overexpression of VEGF has been reported in up to 50% of iCCA and 60% of eCCA [124]. However, data on antitumorigenic effects of several VEGF inhibitors, such as bevacizumab and sorafenib, were rather disappointing [139,140,141,142,143,144,145]. Recently, the multikinase inhibitor regorafenib showed promising efficiency in two phase II trials [146,147]. Based on these encouraging results, Demols et al. conducted a multicenter phase II RCT for patients with nonresectable or metastatic biliary tract cancer and progression on first-line chemotherapy, revealing a significant improvement of PFS and tumor control in patients treated with regorafenib [148]. The VEGFR2 inhibitor apatinib is evaluated in an ongoing RCT as well (NCT03609489).

### 5.5. Isocitrate Dehydrogenase-1 and -2

Isocitrate Dehydrogenase 1 or 2 (IDH) catalytic site mutations exclusively occur in iCCA with a percentage of 18–30% [26,29]. IDH mutations contribute to accumulation of oncometabolite intracellular 2-hydroxyglutarate (2-HG), which in turn disrupts several regulatory cellular pathways that are relevant for epigenetic remodeling and DNA repair [30,149,150]. Elevated levels of circulating 2-HG have been measured in IDH1/2 mutant CCA patients, suggesting the role of 2-HG as a surrogate biomarker of IDH mutation status, tumor burden or treatment response [151]. Although there are data proclaiming that IDH mutations do not have any prognostic or therapeutic significance [2,29,110], recent results demonstrated IDH as a possible target for specific therapy of iCCA [150]. In this context, AG-120 (ivosidenib) is known to inhibit IDH1, whereas AG-221 (enasidenib) inhibits IDH2 [105,149]. In 2020, Abou-Alfa et al. published a multicenter phase III RCT including CCA patients with mutated IDH1 and disease progression upon standard chemotherapy, who were randomly assigned to ivodesinib or placebo treatment. Treatment with ivosidenib significantly improved PFS in this population, while effects on overall survival remain unclear. The optimal use will be explored in future trials [150].

### 5.6. KRAS

Mutations of KRAS and aberrant activation of KRAS signaling pathways occur in up to 40% of CCA, with 42% of mutations appearing in eCCA and 22% in iCCA [105,152]. Currently, KRAS cannot be targeted directly, but rather through inhibition of downstream PI3K-AKT-mTOR and Raf-MEK-ERK pathways, both known to play an important role in cell proliferation, growth and angiogenesis. In 2011, Tanios et al. published a multi-institutional phase II study about the MEK 1/2 inhibitor selumetinib as a treatment for patients suffering from metastatic biliary cancer and demonstrated selumetinib as a well-tolerated drug in combination with current treatment strategies [153]. Furthermore, Kim et al. recently started the first prospective randomized trial on MEK inhibitor trametinib in comparison to chemotherapy with 5-fluorouracil or capecitabine in refractory advanced biliary cancer. However, the study had to be paused because of a lack of clinical activity of trametinib therapy [152]. The combination of trametinib with dabrafenib, a specific BRAF inhibitor, resulted in an overall response rate (ORR) of 47% [154]. Another study investigating the effect of dabrafenib and trametinib in patients with solid tumors is ongoing (NCT02465060). Recent trials evaluating the combination of two KRAS pathway inhibitors offer some promising results [109,155].

### 5.7. Immunotherapy

Inflammation and subsequent activation of the immune system play an important role in carcinogenesis and development of multiple tumors. As such, tumor cell-mediated upregulation of immune checkpoint molecules, e.g., programmed cell death protein 1 (PD-1), results in suppressed immune reaction, facilitating the survival and progression of tumor cells. In fact, the PD-1 antibody pembrolizumab has recently been approved for treatment of unresectable or metastatic microsatellite instability (MSI) high or deficient solid tumors. MSI develops upon loss of DNA mismatch repair mechanism, resulting in stronger immune response and increased expression of PD-1 ligand (PD-L1). While the role of MSI in CCA remains inconclusive, a phase II trial of pembrolizumab efficacy in MMR-deficient tumors demonstrated a 100% disease control rate (DCR) with one complete response and three stable diseases in four patients with CCA [156]. The results of the KEYNOTE-028 phase 1b trial (NCT02054808) revealed the safety and efficacy of pembrolizumab in patients with PD-L1-positive advanced biliary tract cancer, while results from the KEYNOTE-158 phase II trial showed a 40.9% ORR in a subset of 22 patients with MSI high CCA treated by pembrolizumab [157]. Recently the TOPAZ-1 study was able to demonstrate a clinical meaningful OS benefit by adding durvalumab, a PD-L1 inhibitor, to a standard chemotherapy consisting of gemcitabine plus cisplatin (Astra-Zeneca press release October 25, 2021). Further phase-III trials (e.g., Keynote-966, NCT04003636) investigating the role of immunotherapy in CCA are ongoing.

## 6. Conclusions

Cholangiocarcinoma is an anatomically distinct and genetically heterogeneous tumor, with rising incidence worldwide, dismal prognosis and highly limited therapy options. The vast spectrum of genomic alterations differs greatly between each CCA subtype and offers the opportunity for development of prognostic and predictive biomarkers, as well as novel therapeutic strategies. Preoperative identification of prognostic biomarkers may help to identify patients who would profit the most from radical surgical resection, while preventing the risk of postoperative complications and chemotherapy delay. Furthermore, serum and tissue biomarkers could help to identify the subgroup of patients with high recurrence risk, thus necessitating closer follow-up or prolonged chemotherapy regimen. Although numerous options have been developed over the last years, including miRNAs, lncRNAs, SNPs and various signaling molecules from both patients’ serum and tumor tissue, CA19-9 and CEA remain the most applied prognostic biomarkers. In either case, palliative or curative intent, development of custom and patient-directed therapies based on unique genetic alterations will increase the therapeutic effectiveness and consecutively patients’ survival rates. So far, emerging targeted therapies with promising effect include FGFR inhibitors and IDH1/2 inhibitors, as well as immunotherapies. While several other biomarkers and clinical studies have shown promising results, validation in larger patient cohorts and international trials is necessary and has already been initiated to a certain extent. Over the next years, results from multicentric RCTs may fundamentally impact the diagnostic and therapeutic management of CCA and will hopefully improve patient outcomes.

**Table 1 cancers-14-01026-t001:** Serum biomarkers associated with prognosis in CCA.

Name	Occurrence	Expression	Associated Prognostic Value	Reference
Proteins/Cytokines				
CA19-9	CCA (all subtypes)	Increased	OS	[31,32,33,34]
CEA	mostly iCCA, but also all CCA subtypes	Increased	OS	[36,38,39,40,41,42]
CYFRA	iCCA, gallbladder cancer	Increased	OS	[43,158]
Osteopontin	CCA (all subtypes)	Increased	OS	[45]
	iCCA	Low level of circulating osteopontin/volume; Decreased expression in tumor tissue	OS	[46]
Urokinase plasminogen activator receptor (uPAR)	CCA (all subtypes)	Elevated serum levels; Increased expression in tumor tissue	OS	[48]
2-hydroxyglutarate (2-HG)	iCCA	Elevated serum levels	IDH1/2 mutation status, tumor burden	[151]
Nardilysin (NRDC)	iCCA	Elevated serum levels and mRNA expression in tumor tissue	OS, DFS	[87]
IL-6	iCCA	Elevated serum levels	DFS	[47]
**Circulating Nucleic Acids**				
miR-21	CCA (all subtypes)	Elevated serum levels	OS, clinical staging, metastasis	[159]
miR-192	Liver fluke-associated CCA	Elevated serum levels	OS, lymph node metastasis	[160]
miR-106a	CCA	Decreased serum levels	OS, lymph node metastasis	[161]
miR-26a	CCA	Elevated serum levels	OS, clinical stage, metastasis, differentiation status	[162]
Panel (miR-29, miR-122, miR-155, miR-192	CCA	Elevated serum levels	OS	[56]
**Single-Nucleotide Polymorphisms (SNPs)**				
CXCR1 +860 C>G	pCCA	Heterozygous polymorphism	OS, DFS	[65]
G protein subunit-β 3 (GNB3) 825 C>T	eCCA	Heterozygous polymorphism	OS	[66]
EZH2 rs887569 TT genotype	CCA	Homozygous polymorphism	OS	[163]
NRF2 rs6726395 GG genotype	CCA	Homozygous polymorphism	OS	[164]

**Table 2 cancers-14-01026-t002:** Tumor tissue biomarkers associated with prognosis in CCA.

Name	Occurrence	Expression	Associated Prognostic Value	Reference
Cell Surface Molecules				
CD 155	CCA	Increased	OS, DFS, histological grading, lymph node metastasis	[77]
CD44	Liver fluke-associated CCA	Increased	OS	[78]
CD55, CD97	iCCA	Increased	OS, histological grading, lymph node metastasis, venous invasion	[165]
CD98	CCA	Increased	OS	[166]
**Signaling Molecules, Growth Pathways, Angiogenesis**				
IL-6	iCCA	Increased	OS, DFS	[47]
IL-17	iCCA	Increased peritumoral expression	OS, DFS	[47]
SOCS3	CCA	Low intratumoral expression	OS, lymph node metastasis, postoperative disease recurrence	[80]
Tumor necrosis factor α-induced protein 3 (TNFAIP3 or A20)	CCA	Increased intratumoral expression	OS, lymph node metastasis, postoperative disease recurrence	[80]
RNF43	iCCA	Low intratumoral expression	OS	[81]
LIM and SH3 protein 1 (LASP-1)	CCA	Increased intratumoral expression	OS, tumor size, histological differentiation, lymph node metastasis, TNM stage	[82]
B7-H4	CCA	Increased	OS, histological differentiation, lymph node metastasis, staging, early recurrence of tumor	[83]
Hepatoma-derived growth factor (HDGF)	iCCA	Increased	OS, lymph node metastasis, TNM stage	[84]
Ki-67, p73	pCCA	Increased	OS, TNM stage	[85]
Sex-determining region Y-box 4 (SOX4)	iCCA	Increased	OS	[86]
Sex-determining region Y-box 9 (SOX9)	iCCA	Increased	OS	[86]
KRAS	CCA	Increased	OS	[167]
TP53	CCA	Increased	OS	[167]
ARID1A	CCA, mostly fluke-associated iCCA	Decreased	OS	[71,72]
	iCCA	Increased	OS, recurrence rate	[73]
EGFR, MUC1, MUC4, fascin	CCA	Increased	OS	Metanalysis by [74]
VEGF, COX-2, GLUT-1, cyclin D1, Ki67	eCCA	Increased	OS	Metanalysis by [75]
p16, p27, E-cadherin	eCCA	Increased	OS	Metanalysis by [75]
c-MET	CCA	Increased	OS, DFS	[168]
DKK1	iCCA, pCCA	Increased	OS, lymph-node metastasis	[169,170]
BAP1	CCA	Retained expression	OS, DFS	[25,169,170]
		Loss of expression	Trend towards improved OS, histological differentiation, lymph-node metastasis	
PBRM1	CCA	Retained expression	OS, DFS	[29,171]
Mucins				
MUC5AC	Liver fluke-associated iCCA, iCCA	Increased	OS, lymph node metastasis, TNM stage, tumor size	[88,89]
MUC4	CCA	Increased	OS	[172]
MUC16	iCCA	Increased	OS	[173]
**Tumor Stroma and Microenvironment**				
Epithelial cell adhesion molecule (EpCAM)	iCCA	Increased expression in peritumoral stroma	OS, DFS	[90]
Lysil oxidase-like 2 (LOXL2)	iCCA	Increased expression in peritumoral stroma	OS, DFS	[91]
Matrix metalloproteinase -9 (MMP-9)	pCCA	Increased tissue expression	OS	[92]
Matrix metalloproteinase -11 (MMP-11)	CCA	Increased tissue expression	OS	[93]
**Non-Coding RNA**				
lncRNA H19	CCA	Increased tissue expression	OS, DFS, tumor size, TNM stage	[95]
lncRNA-PANDRA	CCA	Increased tissue expression	OS, DFS, lymph node metastasis, TNM stage	[96]
Panel (miR-675-5p, miR-652-3p and miR-338-3p)	iCCA	Overexpression	OS, DFS	[97]
miR-29a	CCA	Overexpression	OS, lymph node metastasis, histological differentiation, clinical staging	[98]
miR-21	Liver fluke-associated iCCA	Overexpression	OS, lymph-node metastasis	[174]
miR-92b	CCA	Overexpression	OS	[175]
miR-34a	eCCA	Decreased expression	OS, lymph-node metastasis, clinical stage	[176]
miR-181a	CCA	Overexpression	OS	[177]
miR-191	iCCA	Overexpression	OS, DFS	[178]
miR-203, miR-373	CCA	Decreased expression	OS, DFS	[178,179]
miR-221	eCCA	Overexpression	DFS	[180]

## Figures and Tables

**Figure 1 cancers-14-01026-f001:**
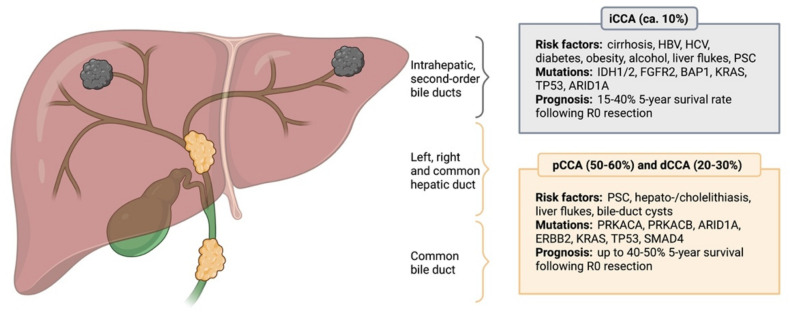
Anatomical classification of cholangiocarcinoma. CCA is anatomically divided into intrahepatic (iCCA), perihillar (pCCA) and distal (dCCA) cholangiocarcinoma, with pCCA and dCCA being summarized as extrahepatic cholangiocarcinoma (eCCA). Different CCA subtypes possess distinct molecular aberrations and differ in terms of their etiology, while certain risk factors and genetic mutations are not subtype-specific. The most common risk factors and prevailing genetic alterations are presented. *HBV:* Hepatitis B virus; *HCV:* Hepatitis C virus; *PSC:* Primary sclerosing cholangitis; *IDH1/2:* Isocitrate dehydrogenase 1/2; *FGFR2:* Fibroblast growth factor receptor 2; *BAP1:* BRCA1 associated protein 1; *KRAS:* Kirsten rat sarcoma virus; *TP53:* Tumor suppressor protein 53; *ARID1A:* AT-rich interactive domain-containing protein 1A; *PRKACA:* Protein kinase cAMP-activated catalytic subunit alpha; *PRKACB:* Protein kinase cAMP-activated catalytic subunit beta; *ERBB2:* Erb-B2 receptor tyrosine kinase 2; *SMAD4:* Mothers against decapentaplegic homolog 4.

**Figure 2 cancers-14-01026-f002:**
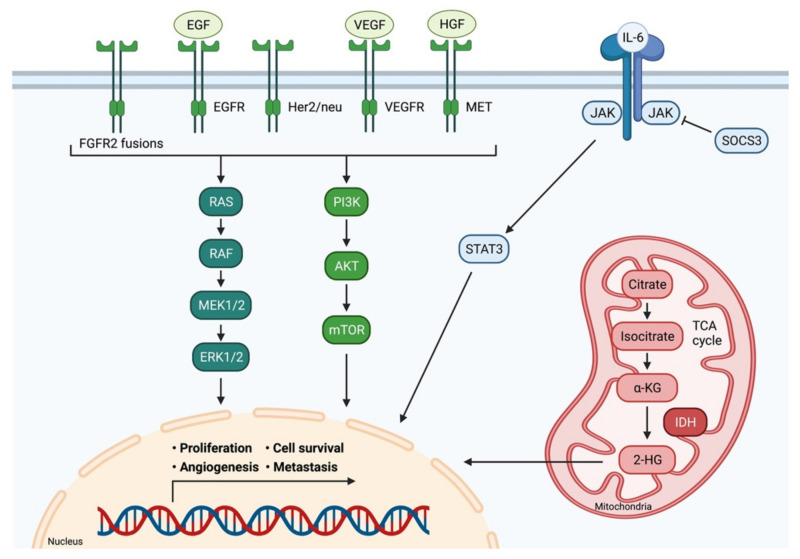
Signaling pathways in cholangiocarcinoma. Multiple signaling pathways are involved in the development and progression of CCA. Receptor tyrosine kinases activate the RAS-MAPK pathway and the PI3K-AKT pathway. IL-6 induces the JAK/STAT signaling pathway. Consequently, these pathways impact important cellular processes, such as cell proliferation, differentiation, survival, and angiogenesis. IDH 1/2 mutations lead to the accumulation of the oncometabolite intracellular 2-hydroxyglutarate (2-HG). Adapted from [30]. *FGFR2:* Fibroblast growth factor receptor 2; *EGF:* Epidermal growth factor; *EGFR:* Epidermal growth factor receptor; *Her2/neu:* Human epidermal growth factor receptor 2; *VEGF:* Vascular endothelial growth factor; *VEGFR:* Vascular endothelial growth factor receptor; *HGF:* Hepatocyte growth factor; *MET:* C-met-encoded receptor for hepatocyte growth factor; *IL-6:* Interleukin-6; *JAK:* Janus kinase; *SOCS3:* Suppressor of cytokine signaling 3; STAT3: Signal transducer and activator of transcription protein; *RAS:* Rat sarcoma; *RAF:* Rat fibrosarcoma; *MEK1/2:* Mitogen-activated protein kinase kinase; *ERK1/2:* Extracellular signal-regulated kinase 1/2; *PI3K:* Phosphatidylinositol 3 kinase; *AKT:* Protein kinase B; *mTOR:* Mammalian target of rapamycin; *α-KG:* α-Ketoglutaric acid; *2-HG:* 2- hydroxyglutarate; *IDH:* Isocitrate dehydrogenase; *TCA cycle:* Citric acid cycle.

## Data Availability

No new data were created or analyzed in this study. Data sharing is not applicable to this article.

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
