# Peer review of "Prognostic and Predictive Molecular Markers in Cholangiocarcinoma"

_cancers, 2022, doi:10.3390/cancers14041026_

Round 1

Reviewer 1 Report

In their detailed and well written review entitled "Prognostic and predictive molecular markers in cholangiocarcinoma", Pavicevic and coll. provided a comprehensive and detailed summary of current evidence concerning molecular markers of Cholangiocarcinoma

Considering its increasing incidence and taking into account its aggressive behaviour, there is an urgent need to identify specific markers that could aid to enhance early diagnosis and predict tumor biology, in order to enhance a better treatment and prognosis.

The paper is generally well written, with appropriate english style.

I have few suggestions:

1) Describe the curative option of liver transplantation in the introduction section, as reported in a recently published review focusing on surgical approaches to p-CCA and i-CCA: 10.3390/cancers13153657

2) Please split the tables in order to enhance their readability

3) Better discuss the problem of adequate sampling for tissue biomarkers utilization

4) Table 3 could be provided as a supplementary table (and splitted for each drug class), as it does not deal with the main paper purpose and is SEVEN pages long!

5) Please review the references style according to journal guidelines (i.e.: include DOIs)

Best regards

Reviewer 2 Report

Thank you for giving me a chance to review a manuscript entitled ”Prognostic and Predictive Molecular Markers in Cholangiocarcinoma”.

This paper would be well written, however, I would like to request two points which should be revised.

In my opinion, serum biomarkers associated with prognosis in CCA would be the key to conduct a prospective study. In this context, Table 1 seems insufficient. Authors should search systematically using public database once more again. At least, 2-HG (Borger et al. Cllin Can Res 2014) and NRDC (Yoh et al. Clin Can Res 2019) should be included in Table 1. Authors already emphasize the importance of IDH mutation in iCCA and EMT process.

Meanwhile, 18F-FDG-PET may surrogate metabolic activity in the CCA. 8F-FDG-PET may play as molecular markers (e.g., surrogate for GLUT-1, HIF-1α, KRAS, Ikeno et al. J Transl Med 2018)

Round 2

Reviewer 1 Report

The Authors properly addressed all the concerns that were raised in the previous report

The revised manuscript is suitable for publication